# Machine Learning-Based Analysis of Differentially Expressed Genes in the Muscle Transcriptome Between Beef Cattle and Dairy Cattle

**DOI:** 10.3390/ijms26115046

**Published:** 2025-05-23

**Authors:** Shuai Li, Yaqiang Guo, Chenxi Huo, Lin Zhu, Caixia Shi, Risu Na, Mingjuan Gu, Wenguang Zhang

**Affiliations:** 1College of Animal Science, Inner Mongolia Agricultural University, Hohhot 010010, China; lishuai@emails.imau.edu.cn (S.L.); gggyaqiang@163.com (Y.G.); 15660097986@163.com (C.H.); zhulinynacxhs@163.com (L.Z.); shicx98@163.com (C.S.); narisu@swu.edu.cn (R.N.); 2Inner Mongolia Engineering Research Center of Genomic Big Data for Agriculture, Hohhot 010010, China; 3College of Life Science, Inner Mongolia Agricultural University, Hohhot 010010, China

**Keywords:** muscle, transcriptome, machine learning, SHAP

## Abstract

Muscle is a crucial component of cattle, playing a vital role in determining the final quality of beef. This study aimed to identify candidate genes associated with muscle growth and lipid metabolism in beef and dairy cattle by utilizing the public database of the National Center for Biotechnology Information (NCBI) to download bovine muscle transcriptome data. Through differential expression analysis, weighted gene co-expression network analysis (WGCNA), and SHapley Additive exPlanation (SHAP) explains machine learning models, we integrated and screened for relevant genes. The results showed a total of 2588 differentially expressed genes (DEGs), with 933 upregulated and 1655 downregulated in beef cattle compared to dairy cattle. In the WGCNA, the purple, black, green, red, brown, and blue modules were identified as significant modules. Based on the results of five different machine learning models, the Adaptive Boosting (AdaBoost) model demonstrated superior classification performance (accuracy = 0.84) compared to the other four models and was therefore selected as the optimal model. SHAP analysis was then employed to interpret the results, yielding the top 500 SHAP genes. In combination with DEGs and WGCNA, a total of 117 genes were identified. Subsequent functional enrichment analysis of these 117 genes revealed significant enrichment in pathways such as lipoprotein metabolic process, muscle contraction, and cytoskeleton in muscle cells, followed by interaction network analysis of genes and pathways. Ultimately, the *APOA1*, *ACTB*, *S1PR1*, *PKLR*, and *SLC27A6* genes were identified as potential key regulators of lipid metabolism and muscle growth in beef and dairy cattle. In summary, this study provides a feasible method for handling large-scale transcriptome data and lays a foundation for future research on meat quality and improving the economic benefits of Holstein cattle.

## 1. Introduction

Muscle tissue constitutes an indispensable physiological component in animals, playing critical roles in locomotion, metabolic homeostasis, and developmental regulation, while also serving as a vital economic trait in livestock production. Beef, renowned for its high nutritional value, represents a major component of global dietary protein intake and ranks among the most widely consumed meats worldwide. Over the past decade, China’s beef production has shown a steady increase, mirroring global trends [1]. Transcriptomics has revolutionized our understanding of gene expression dynamics by enabling genome-wide quantification of RNA molecules. The advent of RNA sequencing (RNA-seq) technology has empowered researchers to dissect complex biological processes across species and tissues, including alternative splicing, non-coding RNA regulation, and isoform-specific expression [2,3]. However, advancements in high-throughput sequencing platforms have precipitated an exponential growth in RNA-seq data volume, posing significant computational challenges. Since the debut of RNA-seq, hundreds of analytical tools have been developed [4]. For instance, Spliced Transcripts Alignment to a Reference (STAR, v2.1.3) software, designed for reference-based alignment, demonstrated robust performance when benchmarked against the ENCODE transcriptomic dataset encompassing 480 billion reads [5]. For organisms lacking reference genomes, de novo transcriptome assembly tools like Trinity have proven indispensable, reconstructing full-length transcripts from short-read data with high fidelity [6], while alignment optimizers such as HISAT2 [7] have enhanced mapping efficiency. These technological advancements have catalyzed research in non-model species, including livestock. Transcriptomic investigations of bovine (*Bos taurus*) muscle development and metabolic adaptation have uncovered critical pathways, such as lipid metabolism (*FABP4*, *ADIPOQ*) and myogenesis (*MYH1*, *ACTA1*), providing mechanistic insights into breed-specific traits [8,9,10].

Despite these advancements, the analysis of large-scale RNA-seq datasets necessitates sophisticated computational strategies. Recent progress in high-performance computing (HPC) and machine learning (ML) has driven the widespread adoption of artificial intelligence (AI) in genomics and animal genetic breeding [11]. In this context, ML methods have demonstrated superior performance over conventional analytical approaches in processing bulk RNA-seq data and identifying candidate gene subsets for phenotypic prediction or classification [12,13,14]. Previous studies have validated the utility of ML frameworks in livestock genomics. For instance, eXtreme Gradient Boosting (XGBoost) has been successfully applied to predict phenotypic traits in livestock by integrating RNA-seq data with genomic variants, achieving high accuracy in identifying genes associated with feed efficiency and intramuscular fat deposition [15]. Yao et al. [16] demonstrated that Random Forest (RF) effectively identifies additive predictors linked to bovine feed efficiency, while Support Vector Machine (SVM) has proven robust for genomic prediction of dairy cattle feed efficiency [17]. Furthermore, Guo et al. [18] conducted cross-species transcriptomic analyses of muscle tissues in cattle, sheep, and swine using Seq2Fun, identifying Adaptive Boosting (AdaBoost) as the optimal model for multi-species phenotypic prediction.

This study employs five machine learning models, interpreted through SHAP, and integrates differential analysis and WGCNA to conduct a comparative transcriptomic analysis of the muscle tissues of beef and dairy cattle. By integrating multiple methods, we have identified candidate genes related to muscle growth and lipid metabolism in beef and dairy cattle. Additionally, this approach enhances production efficiency in Holstein cattle, while the integration of machine learning methodologies provides a new framework for analyzing large-scale transcriptomic data in future studies.

## 2. Results

### 2.1. Data Quality Control and Differential Analysis

Through Seq2Fun processing of transcriptome data, a total of 22,300 genes were obtained. After filtering and normalization, 12,382 genes were obtained. Batch effects across samples from different datasets were adjusted using the ComBat method, followed by cluster analysis, which revealed distinct clustering patterns between beef cattle and dairy cattle breeds (Figure 1A). These well-segregated clusters were deemed suitable for subsequent analytical workflows. Differential expression analysis conducted via EdgeR identified 2588 differentially expressed genes (DEGs), comprising 933 upregulated and 1655 downregulated transcripts (Figure 1B).

### 2.2. Weighted Gene Co-Expression Network Analysis (WGCNA) of the Muscle Transcriptome

To enhance the accuracy of network construction, a final set of 9126 genes was retained after filtration. When a power value of 14 was selected, a high correlation coefficient and elevated average network connectivity were observed (Figure 2A,B). A weighted gene co-expression network model was subsequently constructed, identifying 14 distinct modules in the merged dynamic analysis (Figure 2C). The grey module, representing unassigned genes with no biological relevance, was excluded from further interpretation. Among the modules, the purple module contained the largest number of genes (2021), followed by the brown module (1943 genes), while the dark green module exhibited the smallest gene count (83 genes). Modules significantly associated with phenotypic traits were screened using thresholds of absolute correlation coefficient ≥ 0.3 and *p* < 0.05. The purple, black, green, red, brown, and blue modules were identified as significant (correlation coefficients: 0.79, 0.77, 0.77, 0.72, 0.71, and 0.63, respectively) and were subsequently employed for downstream analysis (Figure 2D).

### 2.3. Evaluate the Classification Performance of Five Machine Learning Models on Beef Cattle and Cow Cattle

To ensure the reliability of the results, model performance was rigorously evaluated using precision, recall, F1-score, and accuracy metrics. The Adaptive Boosting(AdaBoost) model demonstrated the highest accuracy (0.84) among all tested models (Table 1). For validation, 40 test samples were randomly selected from each group. The AdaBoost model misclassified 3 beef cattle samples as dairy cattle and 10 dairy cattle samples as beef cattle (Figure 3A). In contrast, other models such as Support Vector Classifier (SVC) and Multilayer Perceptron (MLP) achieved perfect prediction for dairy cattle but misclassified over half of the beef cattle samples (Figure 3B,C). The Random Forest(RF) model misclassified 8 beef cattle and 20 dairy cattle samples, while the Support Vector Machine (SVM) model misclassified 12 beef cattle and 11 dairy cattle samples (Figure 3D,E). Collectively, the AdaBoost model outperformed other models in both classification accuracy and predictive consistency, aligning with its superior precision, recall, F1-score, and accuracy metrics. This confirms AdaBoost as the optimal model for the dataset among the five candidates.

To interpret the AdaBoost model, SHapley Additive exPlanation (SHAP) analysis was performed, and the top 500 SHAP-ranked genes were identified. Integration of differential expression analysis and WGCNA revealed 105 overlapping genes shared between these approaches (Figure 3F).

### 2.4. Enrichment Analysis and Network Interaction Relationships

Integration of differential expression analysis, WGCNA, and SHAP-based machine learning interpretation identified 105 overlapping genes. Following gene ID conversion, 117 genes were retained for functional enrichment analysis (Appendix A). Gene Ontology (GO) analysis revealed significant enrichment in biological processes such as lipoprotein metabolic process and positive regulation of phospholipid efflux; molecular functions including long-chain fatty acid transmembrane transporter activity, calmodulin binding, and lipid binding; as well as cellular components such as low-density lipoprotein particle and chylomicron (Figure 4A). KEGG pathway analysis demonstrated that these genes were predominantly enriched in the glucagon signaling pathway, cytoskeleton in muscle cells, and thyroid hormone signaling pathway (Figure 4B). Network analysis of some significantly enriched pathways revealed that the genes *APOA1*, *ACTB*, *S1PR1*, *PKLR*, and *SLC27A6* are involved in the regulation of lipid metabolism and muscle growth in beef and dairy cattle, suggesting that these five genes may be key genes involved in muscle growth and development and lipid metabolism. In addition, myosin heavy chain (MYH) genes, including *MYH1*, *MYH2*, *MYH3*, *MYH4*, *MYH8*, and *MYH11*, were found to be significantly enriched in pathways related to muscle contraction and cytoskeleton in muscle cells (Figure 4C).

### 2.5. Analysis of Protein–Protein Interaction Network of Candidate Genes

The protein–protein interaction (PPI) network revealed functional interactions between *APOA1* and proteins including APOE, APOC3, and APOA2. *APOA1* and APOA2, members of the apolipoprotein family, stabilize high-density lipoprotein (HDL) structure through lipid binding and modulate HDL metabolism. APOE, the primary apolipoprotein in chylomicrons, binds to low-density lipoprotein receptor-related proteins in peripheral tissues (e.g., adipose tissue and muscle), facilitating lipoprotein lipase (LPL)-mediated hydrolysis of triglyceride-rich lipoprotein components to release free fatty acids for tissue utilization. APOC3 inhibits LPL activity, thereby suppressing lipolysis(Figure 5A). *ACTB* exhibited interactions with PPARG, RHOA, PFN1, and CFL1. PPARG encodes peroxisome proliferator-activated receptor gamma (PPAR-γ), a master regulator of adipocyte differentiation. PFN1 is annotated in Gene Ontology (GO) as participating in actin binding and cytoskeletal organization, while RHOA and CFL1 are similarly implicated in actin dynamics and cytoskeletal remodeling(Figure 5B). *S1PR1*, a regulator of adipogenesis, was linked to KLF2 in the PPI network. KLF2 modulates mammalian development, including adipogenesis, suggesting its role in enhancing *S1PR1*-mediated muscle growth and lipid metabolism(Figure 5C). *PKLR* (pyruvate kinase) plays a pivotal role in glycolysis. Within its interaction network, LDHA (abundant in skeletal muscle) is involved in glycolytic and pyruvate metabolic regulation, while PCK1 serves as a key hub for gluconeogenesis and lipid biosynthesis. Proteins such as GPI, ENO4, and LDHB were also associated with glycolysis/gluconeogenesis pathways(Figure 5D). For *SLC27A6*, the PPI network highlighted FABP3—a member of the intracellular fatty acid-binding protein (FABP) family—which mediates long-chain fatty acid uptake, intracellular metabolism, and cellular proliferation. PLIN1, essential for adipocyte differentiation, regulates lipid metabolism, and ACSL1 encodes an isozyme of the long-chain fatty-acid-CoA ligase family. These enzymes catalyze the conversion of free fatty acids into fatty acyl-CoA esters, critical for lipid biosynthesis and fatty acid degradation(Figure 5E).

Collectively, the genes (*APOA1*, *ACTB*, *S1PR1*, *PKLR*, and *SLC27A6*) and their interacting partners form a complex regulatory network governing lipid metabolism, muscle growth, and energy homeostasis through synergistic molecular mechanisms.

## 3. Discussion

This study integrated 202 publicly available muscle transcriptomic datasets and employed diverse machine learning models for bulk transcriptome analysis. The dataset encompassed five distinct beef cattle breeds, thereby enhancing biological diversity, while the dairy cattle cohort consisted of the commonly raised Holstein breed. By comparatively analyzing muscle tissues between beef and dairy cattle, this research aims to identify key genes influencing muscle development and adipogenesis, with particular emphasis on promoting intramuscular fat deposition to improve meat quality. While prior studies on Holstein cattle predominantly focused on milk production enhancement, this investigation specifically addresses the economic potential of Holstein bulls for premium meat production to maximize their commercial value. Given the limited availability of public data, the dairy cattle sample size was notably smaller than that of beef cattle. To mitigate this class imbalance challenge, we implemented the Synthetic Minority Over-sampling Technique (SMOTE), an algorithmic approach that effectively augments underrepresented samples while preserving critical biological information [19]. In this study, 105 genes obtained through intersection were subjected to ID conversion using the downstream analysis platform ExpressAnalyst provided by Seq2Fun, resulting in 117 annotated gene symbols, indicating a one-to-many mapping scenario. To investigate this, we consulted the user manual of Seq2Fun software and found that each sequencing read is associated with an ID, which is used to search for its corresponding s2f_id in the gene annotation map. In most cases, a single read may be annotated with multiple IDs, and these IDs typically share the same s2f_id. A read might be annotated to multiple different orthologous gene IDs. In such cases, only the orthologous gene ID with the highest frequency of occurrence is retained. Therefore, during the conversion of s2f_ids in this study, all IDs, including both high- and low-frequency ones, were included, resulting in 105 s2f_ids being converted into 117 genes. However, it is certain that the 117 gene names correspond to the *Bos taurus (cow)* genome. This situation could also be due to the presence of different gene nomenclature systems or transcript variants in the annotation database used by the bioinformatics tool. Furthermore, a potential limitation of this study is the lack of experimental validation. However, the discovery of pathways related to lipid metabolism and muscle development enrichment analysis indirectly supports the consistency of the above results with the objectives of this study. Certainly, for the above results, we have initiated single-cell transcriptome sequencing (10× Genomics platform), and subsequent analyses will focus on the cell-type-specific expression patterns of these five genes in the target tissues, aiming to uncover potential mechanisms at the cellular level.

To date, several genes have been identified as playing pivotal roles in muscle growth and fat deposition, including myocyte enhancer factor 2 (*MEF2*), myostatin (*MSTN*), myogenic regulatory factors (*MRFs*), insulin-like growth factors (*IGFs*), peroxisome proliferator-activated receptor gamma (*PPAR-γ*), Forkhead box transcription factor O1 (*FOXO1*), mitogen-activated protein kinase (*MAPK*), *MYH6*, and calcineurin [20,21]. Furthermore, the expression patterns of *FAM13A* across various tissues in adult cattle were analyzed, revealing that subcutaneous adipose tissue exhibited the highest expression levels among all tissues, with the exception of lung tissue. This finding demonstrates that the *FAM13A* gene provides a functional basis in bovine adipocytes to enhance intramuscular fat deposition in beef cattle [22]. The key genes identified in this study include *APOA1*, *ACTB*, *S1PR1*, *PKLR*, and *SLC27A6*. Notably, *APOA1*, previously reported as one of five candidate genes in a differential expression analysis of Jinnan and Simmental cattle, has been shown to potentially regulate intramuscular fat (IMF) deposition by modulating lipid metabolism [23]. Concurrently, both *PLTP* and *APOA1* have been implicated in glycolipid metabolism and lipid transport processes [24]. Furthermore, our interaction network analysis revealed *APOA2*, a gene belonging to the same apolipoprotein family as *APOA1*. As a major component of high-density lipoprotein, *APOA2* plays a critical role in lipid metabolism and obesity-related pathways. Intriguingly, *APOA2* detected across four distinct dairy cattle breeds has emerged as a strong candidate gene for body composition traits in bovine breeding programs [25]. Ketosis has a negative impact on high-producing dairy cows in the early lactation period. This study reported for the first time that the genetic variant g.-572 A > G in the core promoter region of the *APOA1* gene was associated with ketosis in Chinese Holstein cows, and g.-572 A > G may be used as a genetic marker for ketosis prevention [26]. Therefore, *APOA1* may be involved in both intramuscular fat formation and the regulation of ketosis in cows. *ACTB* is a type of actin, and actin exists in both monomeric (G-actin) and polymeric (F-actin) forms, both of which play key functions such as cell motility and contraction [27]. This study also found that the gene is enriched in the cytoskeleton in muscle cells pathway, which is consistent with our findings. Proteomic approaches have revealed that meat tenderness is influenced by structural proteins (*ACTA1*, *ACTG1*, *ACTB*, *MYL1*, and *PFN1*), co-chaperones, heat shock proteins, regulatory proteins, metabolic proteins, and oxidative stress proteins [28]. Additionally, RefFinder demonstrated that *TUB*, *ETFDH*, and *ACTB* were highly stable in bovine milk small extracellular vesicles (sEVs) [29]. Among these, the gene *PPARG*, which interacts with *ACTB*, is a key regulator of adipocyte differentiation. Research has shown that bta-miR-130a regulates milk fat biosynthesis by targeting *PPARG* mRNA. Furthermore, overexpression of bta-miR-130a/b inhibited the expression of adipocyte differentiation-related genes, including *PPARG*, *C/EBPα*, *C/EBPβ*, *FABP4*, *LPIN1*, and *LPL* [30]. Through genome-wide association study (GWAS) and weighted gene co-expression network analysis (WGCNA), the genes *NTMT1*, *FNBP1*, and *S1PR1* were identified as the most critical candidate genes influencing the lifelong productivity of Holstein dairy cattle [31]. Meanwhile, analysis of whole-genome SNP data in Hanwoo cattle revealed two highly homozygous regions, likely under strong and/or recent positive selection, containing five genes (*DPH5*, *OLFM3*, *S1PR1*, *LRRN1*, and *CRBN*) [32]. Enrichment analysis of *PKLR* demonstrated its significant association with the “Cellular response to insulin stimulus” pathway. Studies indicate that *PKLR* is a potential candidate gene for milk production traits in cows. The *PKLR* gene exhibited differential expression across lactation stages in dairy cattle and was found to participate in lipid metabolism via insulin, PI3K-Akt, MAPK, AMPK, mTOR, and PPAR signaling pathways [33,34]. In a swine study, the AC genotype of the *CAST* gene was identified as the most favorable for lean meat rate. The positive effect of this genotype generally increased with each additional C allele of the *MYF6* and *TNNT3* genes, while it decreased with each additional T allele from the *SFRS1* and *PKLR* genes [35]. This suggests that an increase in the T allele of *PKLR* may indirectly lead to enhanced fat deposition. Regarding the candidate gene *SLC27A6*, it is a member of the fatty acid transporter protein (FATP) family, which is involved in the uptake of long-chain fatty acids. It mediates the import of long-chain fatty acids (LCFA) into cells by facilitating their transport across the plasma membrane [36]. A study by Zhang et al. [37] demonstrated that knockdown of the *SLC27A6* gene significantly downregulated the mRNA abundance of genes associated with fatty acid activation (*ACSL4*), oxidation (*CPT1A*), and transport (*CD36*), while upregulating the mRNA levels of genes linked to transcriptional regulation (*PPARG*), diacylglycerol O-acyltransferase 1 (*DGAT1*), fatty acid binding (*FABP3*), and desaturation (*FADS2*). This provides robust evidence for the central role of *SLC27A6* in regulating fatty acid metabolism in bovine mammary epithelial cells (BMECs). Additionally, polymorphisms in *FABP4* and *SLC27A6* have been proposed as markers for selecting cattle that produce milk with lower saturated fatty acid (SFA) and higher unsaturated fatty acid (UFA) concentrations [38]. In Japanese Black cattle, a genome-wide association study (GWAS) identified *SLC27A6* as a candidate gene for fatty acid metabolism. Analysis of variance revealed that the *SLC27A6* K81M variant exhibited a stronger association (*p* = 0.0009) than the most significant SNP identified in GWAS (*p* = 0.0049) [39].

In addition, myosin heavy chains (MyHCs), including *MYH1*, *MYH2*, *MYH3*, *MYH4*, *MYH8*, and *MYH11*, were identified in enrichment analyses as key regulators of the “Muscle contraction” and “Cytoskeleton in muscle cells” pathways. MyHC isoforms and muscle fiber cross-sectional area are critical variables in livestock growth, muscle biology, and meat science research [40]. Muscle fiber types are categorized into Type I and Type II (subtypes IIa, IIx, and IIb) [41,42], with *MYH1*, *MYH2*, and *MYH4* belonging to the Type II group. Studies have shown that *MYH3* exhibits high expression levels in muscle tissues, with its expression peaking during the early stages of myoblast differentiation [43]. These genes play essential roles in bovine muscle fiber composition and muscle development.

## 4. Materials and Methods

### 4.1. Source of Transcriptome Data

All data in this study were sourced from the high-throughput sequencing data repository SRA (Sequence Read Archive) within the public database of the National Center for Biotechnology Information (NCBI, https://www.ncbi.nlm.nih.gov/, accessed on 7 January 2025), totaling 202 muscle transcriptome datasets (Appendix A). Among these, there were 113 beef cattle transcriptome samples, including Angus, Simmental, Hanwoo, Nelore, and Hereford; and 89 dairy cattle transcriptome samples, all of which were Holstein cows.

### 4.2. Processing and Quality Control of Transcriptome Data

In this study, raw sequencing data were retrieved and downloaded from the SRA database to a local server. The SRA Toolkit (v3.12.0) software was used with default parameters to convert data from SRA format to Fastq format. The Seq2Fun (v2.0.5) software was employed for the analysis of transcriptome Fastq files, utilizing the mammalian database (mammals_v2.0.fmi, mammals_annotation_v2.0.txt) provided by the official website for sequence alignment and annotation to achieve automatic data quality control. For large-scale transcriptome data, the Seq2Fun software can complete data processing quickly and efficiently while conserving computational resources. Finally, the SVA package in R (v4.4.1) was used to eliminate batch effects from different datasets through the Combat method.

### 4.3. Differential Analysis of Muscle Transcriptome

Downstream transcriptomic analysis was performed using ExpressAnalyst, the integrated analytical platform provided by the Seq2Fun suite, to conduct differential expression profiling. The raw expression matrix was preprocessed by filtering out low-abundance genes (Abundance threshold:5; Variance filter: 15). Data normalization was implemented via log2-counts per million (log2CPM) transformation to stabilize variance across heterogeneous expression ranges. Differential analysis was subsequently executed using EdgeR, with statistically significant differentially expressed genes identified under stringent thresholds of |log2(fold change)| ≥ 0.5 and *p*-value < 0.05. This systematic pipeline ensured robust detection of transcriptional variations while controlling technical biases.

### 4.4. Weighted Gene Co-Expression Network Analysis of Muscle Transcriptome

To investigate the co-expression patterns of genes, the Weighted Gene Co-expression Network Analysis (WGCNA) method was employed using a web platform (https://cloud.oebiotech.com/). Genes with low expression levels (standard deviation ≤0.5) were filtered out to create a scale-free topological network. Soft thresholding power was used to adjust the weights of the adjacency matrix, aiming to make the network as close to scale-free as possible. The appropriate value for constructing the scale-free network was determined when the fitting index ranged between 0.8 and 0.9. Finally, the dynamic tree cutting algorithm was used for module division [44]. In addition, the correlation between module gene expression and traits (GS) was determined using a threshold of absolute correlation coefficient ≥ 0.3 and *p* < 0.05.

### 4.5. Machine Learning and Model Interpretability

The ever-expanding scale and inherent complexity of biological data have driven the increasing application of machine learning in biology to establish potential biological process information and predictive models [45]. In this study, five models were constructed using Jupyter in Anaconda Navigator (v2.6.3) software, including Support Vector Classifier (SVC), Multilayer Perceptron (MLP), Support Vector Machine (SVM), Adaptive Boosting (Adaboost), and Random Forest (RF). The optimal model was selected through the analysis of the transcriptome expression matrix. Each model randomly selected 40 samples from beef and dairy cattle as the test set, with the remaining samples used as the training set. Due to the different sample sizes from various cattle breeds, the Synthetic Minority Over-sampling Technique (SMOTE) was employed to address the issue of data imbalance [46]. Finally, SHapley Additive exPlanation (SHAP) [47] was used to interpret the model and calculate the SHAP values of genes.

### 4.6. Enrichment Analysis and Network Interaction Analysis

Combining the results from the aforementioned differential analysis, WGCNA, and SHAP gene selection, we performed GO and KEGG enrichment analyses on all overlapping genes. Since we used the Seq2Fun tool for processing the raw transcriptome Fastq files, all gene names in this study are derived from the Seq2Fun database’s custom definitions. Therefore, it is necessary to convert these to commonly used gene names. For this purpose, we continue to use the ExpressAnalyst online platform to convert the Seq2Fun custom-defined gene names to those corresponding to *Bos taurus (cow)*. The successfully converted genes were then subjected to enrichment analysis using the DAVID database (https://davidbioinformatics.nih.gov/, accessed on 28 March 2025), with R (v4.4.1) software used for plotting. Finally, Cytoscape (v3.8.2) software was employed to illustrate the interaction relationships between genes and related pathways.

### 4.7. Protein–Protein Interaction Network

Protein–protein interaction network analysis of the identified key genes was conducted and visualized using the STRING database (https://string-db.org). Among them, the confidence score threshold for key regulators is medium confidence (0.400).

## 5. Conclusions

In summary, this study conducted a comprehensive analysis of muscle transcriptomes in beef and dairy cattle through the integrated application of differential expression analysis, WGCNA, and machine learning approaches. Ultimately, the *APOA1*, *ACTB*, *S1PR1*, *PKLR*, and *SLC27A6* genes were found to be involved in the regulation of lipid metabolism and muscle growth in beef and dairy cattle, potentially serving as crucial candidate genes influencing muscle growth and lipid metabolism. This study provides a feasible method for processing large-scale transcriptome data and offers fundamental insights for subsequent research on meat quality improvement and enhancing the economic value of Holstein cattle breeding programs.

## Figures and Tables

**Figure 1 ijms-26-05046-f001:**
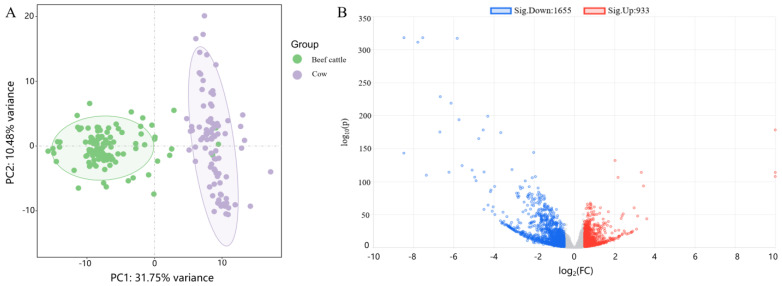
Cluster analysis and differential analysis of the sample group. (**A**) Principal component analysis (PCA) plot of sample group clustering. (**B**) Volcano plot of differential analysis.

**Figure 2 ijms-26-05046-f002:**
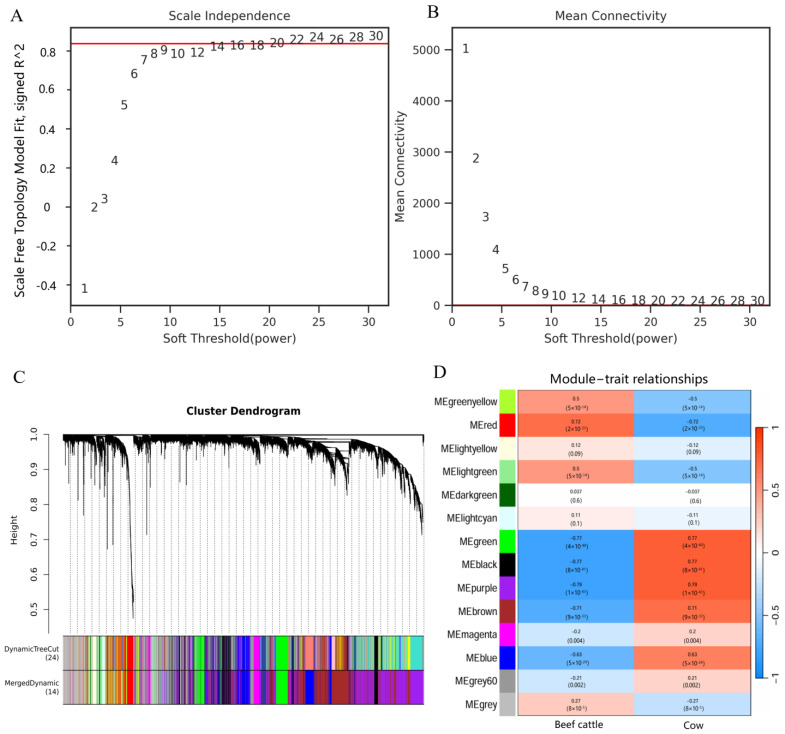
Network parameter construction and module identification in WGCNA. (**A**) Correlation coefficients corresponding to different powers. (**B**) Average node connectivity of networks constructed with different power values. (**C**) The upper part of the figure shows the gene clustering tree constructed after constructing the dissTOM matrix with weighted correlation coefficients; the lower part of the figure shows the distribution of genes in each module. (**D**) Heatmap of module associations.

**Figure 3 ijms-26-05046-f003:**
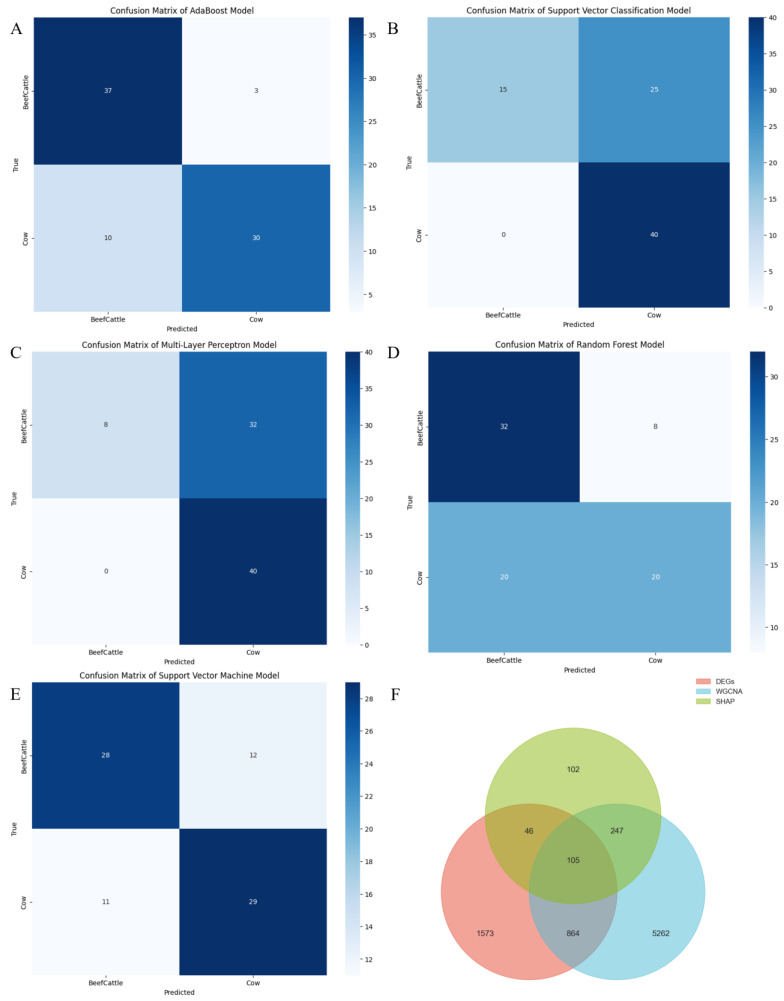
Classification results of different machine learning models for beef cattle and dairy cattle. (**A**) Confusion matrix of the AdaBoost model. (**B**) Confusion matrix of the Support Vector Classifier model. (**C**) Confusion matrix of the Multi-Layer Perceptron model. (**D**) Confusion matrix of the Random Forest model. (**E**) Confusion matrix of the Support Vector Machine model. (**F**) Venn diagram of DEGs, WGCNA, and SHAP genes.

**Figure 4 ijms-26-05046-f004:**
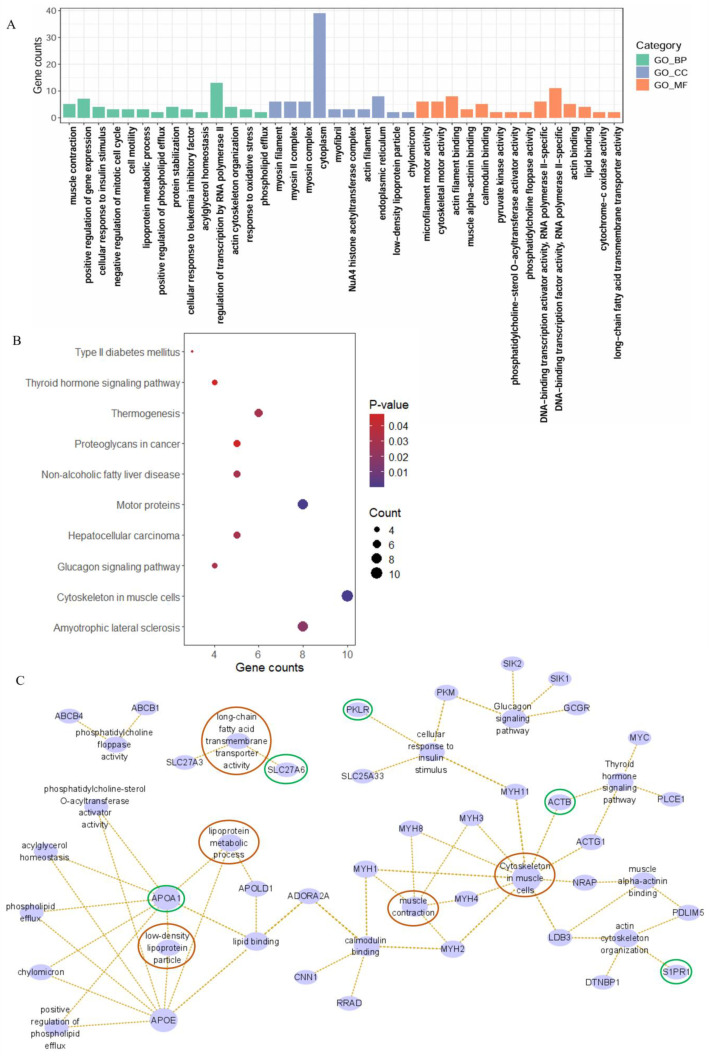
Enrichment analysis. (**A**) GO enrichment analysis of 117 genes. (**B**) KEGG enrichment analysis of 117 genes. (**C**) Interaction network of related pathways and genes. (Red circles represent pathways associated with enrichment analysis, and green circles represent genes associated with that pathway.)

**Figure 5 ijms-26-05046-f005:**
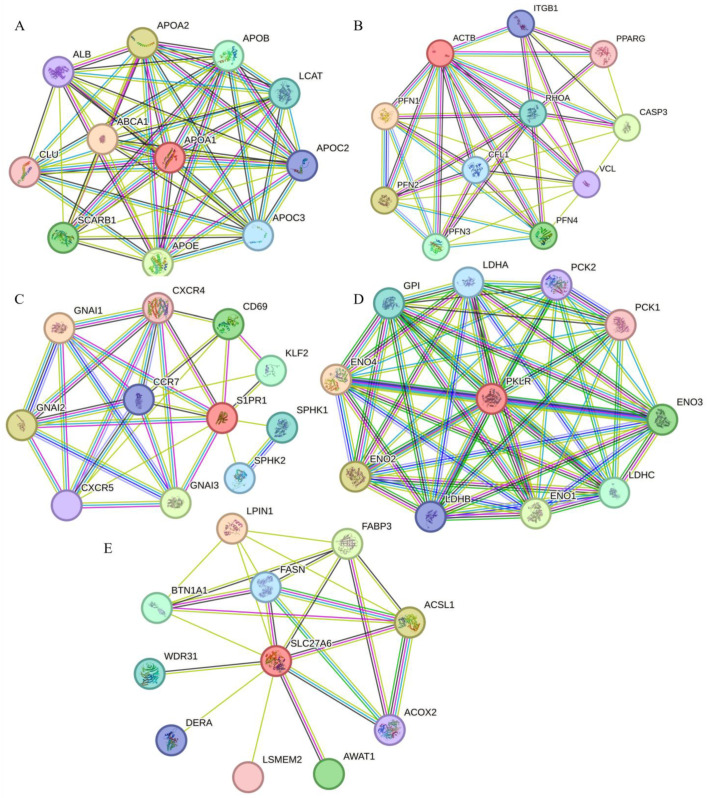
Protein–protein interaction network diagrams of key genes. (**A**) Protein interaction network diagram of *APOA1*. (**B**) Protein interaction network diagram of *ACTB*. (**C**) Protein interaction network diagram of *S1PR1*. (**D**) Protein interaction network diagram of *PKLR*. (**E**) Protein interaction network diagram of *SLC27A6*.

**Table 1 ijms-26-05046-t001:** Precision, recall, F1-scores, and accuracy of different models.

Module	Precision	Recall	F1-Score	Accuracy
AdaBoost	0.85	0.84	0.84	0.84
SVM	0.71	0.71	0.71	0.71
MLP	0.78	0.60	0.52	0.60
SVC	0.81	0.69	0.65	0.69
RF	0.66	0.65	0.64	0.65

## Data Availability

The datasets used in this study were all sourced from the public database NCBI. Specific information about the datasets can be found in Appendix A. This study did not generate original code.

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
