# Peer review of "Machine Learning-Based Analysis of Differentially Expressed Genes in the Muscle Transcriptome Between Beef Cattle and Dairy Cattle"

_ijms, 2025, doi:10.3390/ijms26115046_

Round 1
Reviewer 1 Report
Comments and Suggestions for Authors
Comments and Suggestions:
Title: Machine Learning-Based Analysis of Differentially Expressed Genes in the Muscle Transcriptome between Beef Cattle and Dairy Cattle.
The manuscript by Li et. al., describes about the muscle transcriptomics analysis between beef and dairy cattle using machine learning based algorithms to identify genes associated with muscle growth and lipid metabolism. They used WGCNA and SHAP-explained ML models and identified 2588 DEGs, with 933 up- and 1655 down-regulated genes. After combining DEGs and WGCNA, 117 genes were identified which were enriched in Lipoprotein metabolic process, muscle contraction, and cytoskeleton in muscle cells, with APOA1, ACTB, S1PR1, PKLR, and SLC27A6 emerging as key regulators of lipid metabolism and muscle growth, establishing a foundation for improving meat quality and the economic benefits of Holstein cattle.
The manuscript does not provide novelty, but still some points need to be addressed.
Major Points:
- A systematic flow diagram can be included for clear view of the study.
- Section 2.2: What was the reason for selectin six WGCNA modules? The best module can be selected based on highest coefficient and pvalue and then carry out further analysis.
- Line 140-141: Please clarify how 105 genes after gene ID conversion becomes 117? It should be less than 105 after applying filtering criteria.
- Figure 5: The results of STRING analysis showed five important genes, APOA1, ACTB, S1PR1, PKLR, and SLC27A6. The analysis is very week and only using one type of selection criteria is not enough to select key regulators. The study lacks validation both in in vivo or in vitro.
- Figure 5: What was the confidence score cutoff used for selecting the important key regulators.
- Please add limitations of this study in the conclusion section.
Minor Points:
- Please reduce the plagiarism rate.
- Appendix 1: Please include the log2FoldChange and pvalue columns.
Reviewer 2 Report
Comments and Suggestions for Authors
This manuscript presents a comprehensive multi-method transcriptomic study comparing muscle tissues between beef and dairy cattle. By integrating differential gene expression analysis, Weighted Gene Co-expression Network Analysis (WGCNA), and machine learning (ML) with SHAP interpretability, the study identifies candidate genes involved in muscle development and lipid metabolism. The topic is relevant to livestock genomics and bioinformatics, and the approach is timely given the increasing availability of high-throughput sequencing data and computational tools.
However, despite the novelty and ambition of the study, the manuscript has some issues that must be addressed before being considered for publication.
- Critical biological variables such as sex, age, and breed were not clearly accounted for during DEG analysis. These factors are known to influence transcriptomic profiles and may confound the observed differences. Please clarify whether these were controlled, corrected for, or at least assessed.
- What was the threshold that the raw expression matrix was preprocessed by filtering out low-abundance genes.
- Five models of Machine Learning were used. How about the predicted genes associated with muscle growth and development of each method?
